# *Toxoplasma gondii* phosphatidylserine flippase complex ATP2B-CDC50.4 critically participates in microneme exocytosis

**Hugo Bisio¤, Aarti Krishnan, Jean-Baptiste Marq, Dominique Soldati-Favre\***

Department of Microbiology and Molecular Medicine, CMU, Faculty of Medicine, University of Geneva, Geneva, Switzerland

¤ Current address: Information Génomique & Structurale, CNRS, Aix-Marseille Université, Marseille, France.
* dominique.soldati-favre@unige.ch

**Data Availability Statement:** All relevant data are within the manuscript and its Supporting Information files.

## Abstract

Regulated microneme secretion governs motility, host cell invasion and egress in the obligate intracellular apicomplexans. Intracellular calcium oscillations and phospholipid dynamics critically regulate microneme exocytosis. Despite its importance for the lytic cycle of these parasites, molecular mechanistic details about exocytosis are still missing. Some members of the P4-ATPases act as flippases, changing the phospholipid distribution by translocation from the outer to the inner leaflet of the membrane. Here, the localization and function of the repertoire of P4-ATPases was investigated across the lytic cycle of *Toxoplasma gondii*. Of relevance, ATP2B and the non-catalytic subunit cell division control protein 50.4 (CDC50.4) form a stable heterocomplex at the parasite plasma membrane, essential for microneme exocytosis. This complex is responsible for flipping phosphatidylserine, which presumably acts as a lipid mediator for organelle fusion with the plasma membrane. Overall, this study points toward the importance of phosphatidylserine asymmetric distribution at the plasma membrane for microneme exocytosis.

## Author summary

Biological membranes display diverse functions, including membrane fusion, which are conferred by a defined composition and organization of proteins and lipids. Apicomplexan parasites possess specialized secretory organelles (micronemes), implicated in motility, invasion and egress from host cells. Microneme exocytosis is already known to depend on phosphatidic acid for its fusion with the plasma membrane. Here we identify a type P4-ATPase and its CDC50 chaperone (ATP2B-CDC50.4) that act as a flippase and contribute to the enrichment of phosphatidylserine (PS) in the inner leaflet of the parasite plasma membrane. The disruption of PS asymmetric distribution at the plasma membrane impacts microneme exocytosis. Overall, our results shed light on the importance of membrane homeostasis and lipid composition in controlling microneme secretion.

**Funding:** This work was supported by the Swiss National Foundation to D.S.-F. (FN3100A0-116722) and by the Scientific & Technological Cooperation Programme Switzerland-Rio de Janeiro (IZRJZ3_164183). H.B. is the recipient of a Swiss Government Excellence Scholarship with Uruguay. The funders had no role in study design, data collection and analysis, decision to publish, or preparation of the manuscript.

**Competing interests:** The authors have declared that no competing interests exist

## Introduction

The phylum of Apicomplexa encompasses a diverse group of obligate intracellular protozoan parasites responsible for severe diseases in animals and humans. *Plasmodium spp.*, the etiological agent of malaria, account for half a million deaths per year (World Malaria Report 2017, WHO). *Cryptosporidium* is one of the most important agents causing severe diarrhea in children [1]. Relevant for the farming industry, *Eimeria* and *Theileria* are responsible for a considerable economic burden [2,3]. *Toxoplasma gondii* is the most ubiquitous member of the phylum, capable of infecting humans and animals. The lytic cycle of apicomplexan parasites is tightly controlled to ensure parasite survival and dissemination [4,5]. Underpinning several steps of the lytic cycle is the release of apical secretory organelles called micronemes that are conserved in all motile and invasive stages of apicomplexans [4]. The micronemes secrete adhesins, perforins and proteases that allow gliding, invasion and egress of the parasites [6,7]. In a simplified model, microneme exocytosis is regulated and initiated by the production of cyclic guanosine monophosphate (cGMP) via a signaling platform composed of an atypical guanylate cyclase (GC) fused to a P type-IV ATPase (P4-ATPase) and associated to CDC50.1 as well as to a unique GC organizer, UGO [8–11]. The level of cGMP is tightly controlled by phosphodiesterases, which differential phosphorylation state upon depletion of the protein cAMP-dependent protein kinase (PKA-C1) indicates that are presumably regulated by cyclic adenosine monophosphate (cAMP) levels [12,13]. In turn, cGMP leads to the activation of the cGMP-dependent protein kinase (PKG) [14] which triggers a signaling cascade that involves the production of inositol-tri-phosphate (IP3) and diacylglycerol (DAG) by phosphoinositide phospholipase C (PI-PLC) [15]. IP3 is believed to mobilize calcium from an unknown intracellular store of the parasite [16] and activate calcium-dependent protein kinases (CDPKs) [17,18]. Both CDPK1 and CDPK3 contribute to microneme exocytosis while CDPK1 additionally extrudes the conoid and activates the actomyosin system [17–19], allowing parasite gliding motility, invasion and egress [20]. On the other hand, DAG is converted into phosphatidic acid (PA) through a reversible reaction catalyzed by the DAG kinase 1 (DGK1) and PA phosphatases (PAPs) [21]. Importantly, several feedback loops are expected to feed into different steps of this signaling pathway.

The endomembrane system of the cell displays diverse functions conferred by a defined composition and organization of proteins and lipids. In particular, the exocytosis of secretory organelles depends on specific phospholipids that select the target membranes and trigger fusion. In apicomplexan parasites, PA acts as an essential lipid mediator for microneme exocytosis [21,22]. PA is produced in the inner leaflet of the plasma membrane and allows the docking of micronemes with the assistance of the acylated pleckstrin-homology domain-containing protein (APH) on the microneme organelle surface [21,23]. Importantly, the asymmetric distribution of phospholipids (PLs) across the plasma membrane is known to generate a physical surface tension that is used to induce membrane curvature, favoring vesicle budding and fusion [24]. These gradients are set and maintained by different groups of proteins including P4-ATPases, which function as flippases and form heterodimeric complexes with cell division control protein 50 (CDC50), that act as cofactors and chaperones (Fig 1A) [25]. P4-ATPases possess ten predicted transmembrane spanning domains with cytosolic domains mediating nucleotide binding (N), phosphorylation (P) and dephosphorylation (A) (Fig 1A) [26]. The P4-ATPases couple the hydrolysis of ATP with inward PLs translocation via a Post-Albers mechanism [25], transitioning between two intermediate states: E1 and E2, with different affinities to substrates. During this process, the transmembrane region remainsstructurally rigid based on its interaction with the CDC50 partner [27]. Among the repertoire of P4-ATPases in apicomplexan parasites, some are essential for parasite survival [28–30] yet little

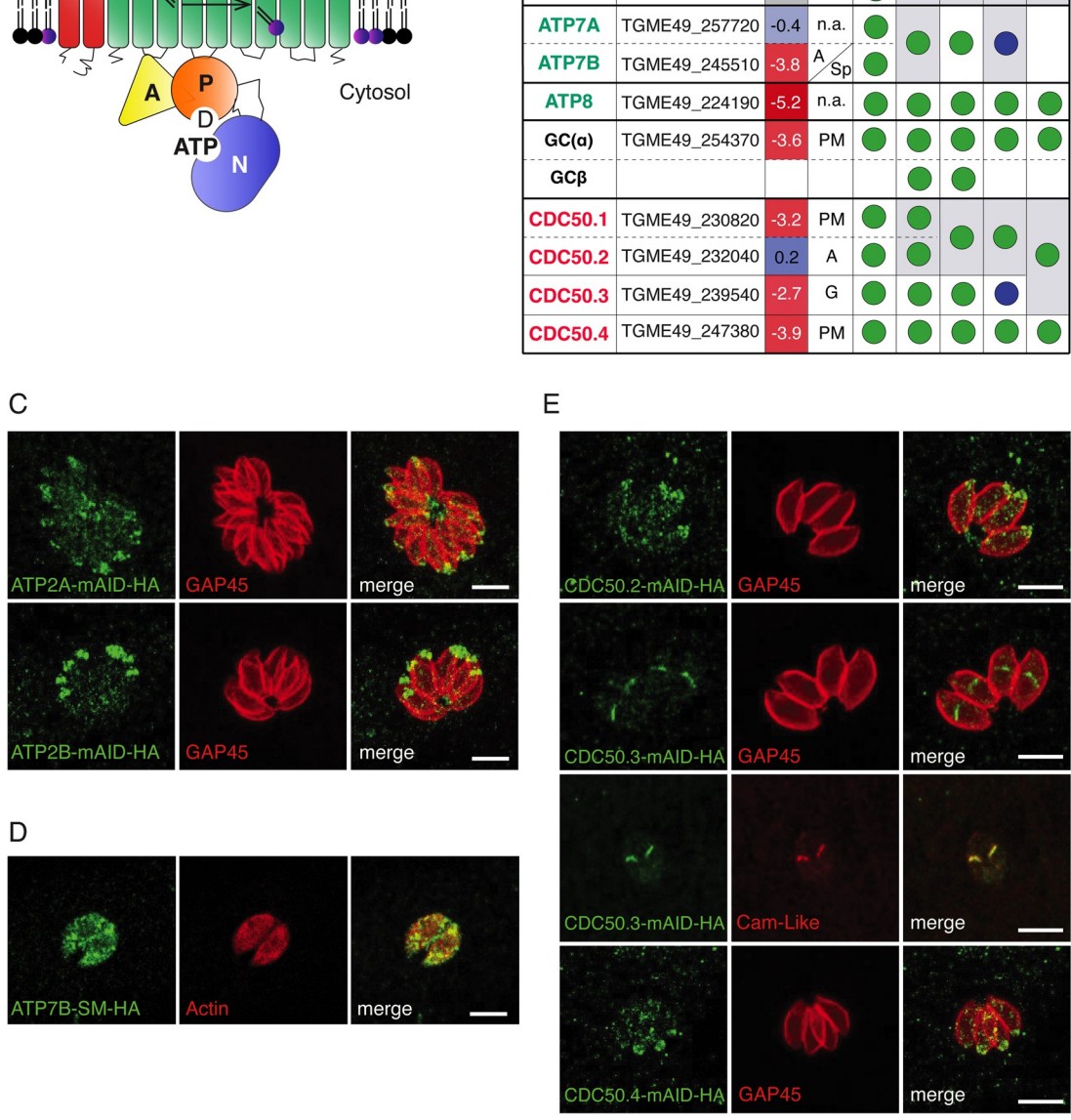

**Fig 1. Members of the Apicomplexa phylum encode a plethora of P4-ATPases and CDC50 cofactor at different parasite locations.** (A) Schematic representation of the domain architecture of the P4-ATPase-CDC50 heterocomplex. (B) Conservation of P4-ATPases and CDC50 cofactors across the Apicomplexa phylum. Blue circle: Absent in *Theileria*. *Eimeria* possesses two genes belonging to the CDC50.1/CDC50.2 subgroups but direct homology could not be deducted by blast analysis. See S1 Fig. Fitness scores associated to gene disruption in *T. gondii* are obtained from [28]. CM: Cyst-forming. PM: plasma membrane. G: Golgi apparatus. A: Apical. S.p: Secretory pathway. N.a: Not assessed. Accession numbers of all putative orthologs genes are included in S1 Table. (C) Indirect immunofluorescence assay (IFA) of intracellular ATP2A-mAID-HA and ATP2B-mAID-HA parasites. GAP45: parasite periphery. (D) IFA of intracellular ATP7B-SM-HA parasites. Actin: Parasite cytosol. (E) IFA of intracellular CDC50.2, CDC50.3 and CDC50.4-mAID-HA parasites. GAP45: parasite pellicle. Cam-Like protein: Golgi apparatus. The scale bars for the immunofluorescence images are 7μM, unless otherwise indicated.

is known about their biological roles and enzymatic functions. Beside the lipid composition, membrane fusion is a universal process that also involves a machinery composed of SNAREs (for "soluble N-Ethylmaleimide-sensitive factor (NSF)-attachment protein receptor") [31]. During fusion, vesicular and target SNAREs assemble into an α-helical trans-SNARE complex that forces the two membranes tightly together [31]. Additionally, this machinery is controlled by C2-containing proteins, like synaptotagmin/ferlin and DOC2, in a calcium-dependent manner [32,33]. The SNARE proteins appear to play pleiotropic functions in *T. gondii* with none identified to date to be uniquely associated to microneme secretion [34]. Contrastingly, both DOC2.1 and Ferlin 1 (FER1) are solely dedicated to microneme secretion in *Plasmodium spp.* and *T. gondii* [35–38]. While DOC2.1 function strictly participates in microneme exocytosis [35], FER1 is involved in microneme proteins trafficking in addition to exocytosis [37], although some of these phenotypes might result from indirect dominant negative effects.

In this study, we address the importance of several P type-IV ATPases and CDC50 chaperones in *T. gondii*. We show that ATP2B forms a heterocomplex with CDC50.4 and acts as an essential flippase to maintain phosphatidylserine (PS) enrichment in the inner leaflet of the parasite plasma membrane. ATP2B and CDC50.4 crucially contribute to microneme exocytosis indicating that PS is a key lipid participating in microneme fusion with the plasma membrane.

## Results

### Identification and localization of the putative flippases and their CDC50 partners in *Toxoplasma gondii*

Sequence homology search in *T. gondii* genome identified six genes predicted to code for P4-ATPases and four genes for CDC50 cofactors (Fig 1B and S1 Table). We utilized the nomenclature already established for *Plasmodium* species in order to name the putative homologues of P4-ATPases [39]. We have chosen this nomenclature, instead of the one presented in a recent overlapping report [30], in order to provide an integrative view of different apicomplexan parasites P4-ATPases and facilitate comparisons. ATP2A and ATP2B appear to be paralogues that are present either as one or two copies across all members of the Apicomplexa phylum. Similarly, ATP7A and ATP7B are putative paralogues found as a pair in the cyst-forming coccidian subgroup of Apicomplexa but absent in *Theileria* and *Cryptosporidia*. Plausible orthologs of *T. gondii* ATP8 and GC are found across the phylum (Fig 1B and S1 Table) with GC being duplicated in the *Plasmodium* and *Eimeria* spp., as previously reported [5] (Fig 1B and S1 Table). The CDC50 protein family is composed of four members in *T. gondii*. CDC50.1, CDC50.2 and CDC50.3 are clustered phylogenetically and may have arisen through gene duplication (S1 Fig). The presence of three individual genes belonging to this group is only found in coccidians (Fig 1B and S1 Table). A single gene assigned to the CDC50.1/2/3 group is present in *Cryptosporidia* while CDC50.3 is absent in *Theileria*. In contrast, CDC50.4 is conserved across the entire phylum (Figs 1B, S1 and S1 Table).

To determine their localization and scrutinize their function, the genes corresponding to the P4-ATPases and CDC50s were C-terminally tagged with 3-HA epitope tags and concomitantly fused to the auxin-inducible degron (mAID) [14] at their endogenous locus via CRISPR-Cas9 genome editing. The resulting mutants were cloned and confirmed by genomic PCR (S2A Fig). Similar to GC and its partner CDC50.1, which have previously been found at the apical cap of the parasites [8], ATP2A, ATP2B, CDC50.2 and CDC50.4 were also found at the apical cap (Fig 1C and 1E). Contrastingly, CDC50.3 localized to the Golgi apparatus (Fig 1E) while ATP7B could only be detected throughout the secretory pathway after tagging with

the spaghetti monster-HA (SM-HA) [40] (Figs 1D and S3B and S3C). ATP8 was refractory to genetic modification, which hampered further investigation.

## *Toxoplasma gondii* CDC50.4 forms promiscuously stable heterocomplexes with ATP2A and ATP2B

Given its conservation across the Apicomplexa, localization and predicted essentiality, we focus our attention on CDC50.4 [28]. In order to identify the complex formed by this protein, we performed immunoprecipitation of CDC50.4 coupled with mass spectrometry analysis (S2D Fig). ATP2A and ATP2B were identified as interacting partners of this protein (Fig 2A). Other proteins identified in pull-down are likely contaminants that correspond to highly abundant proteins or are predicted to be implicated in non-related functions. Moreover, endogenous epitope-tagging of ATP2B-Ty in CDC50.4-mAID-HA confirmed co-localization of the two proteins at the apical tip of the parasite (Fig 2B) and pull-down experiments provided further evidence of their stable association (Fig 2C). Compellingly, depletion of CDC50.4 led to a significant decrease in ATP2B protein level (Figs 2D and S2E). Partial colocalization and downregulation of ATP2A upon depletion of CDC50.4 were also shown by immunofluorescence (Fig 2E). Importantly, double tagging of ATP2A and CDC50.4 rendered a partial miss localization of the complex (Fig 2E), indicating some steric impediment for correct trafficking upon presence of both C-terminal tags. We also observed no decrease in ATP2A-Ty levels upon depletion of CDC50.4 (S2F Fig). The absence of CDC50.4 did not impact on the localization of GC-Ty (Fig 2F and 2G) and its level of expression (Fig 2H) although it shares the same localization as ATP2B but forms a heterocomplex with CDC50.1 [9]. Taken together, these data strongly indicate that CDC50.4 is forming a complex with ATP2A and ATP2B.

## ATP2B-CDC50.4 complex and ATP7B are critical for the survival of *Toxoplasma gondii*

The three P-type IV ATPases (ATP2A, ATP2B and ATP7A) and CDC50s (CDC50.2, CDC50.3 and CDC50.4), were C-terminally fused to the auxin-inducible degron (mAID) at the endogenous locus [14] and efficiently depleted upon addition of 3-indoleacetic acid (IAA) as confirmed by western blot (Fig 3A and 3B). The relative fitness of each knockdown mutant was assessed by its ability to form plaques of lysis on host cell monolayers (Fig 3C and 3D). Parasite lacking ATP2A and CDC50.2 had no apparent fitness defect, whereas loss ATP2B, ATP7B and CDC50.4 in presence of IAA led to significantly smaller plaques compared to parasites grown in absence of IAA (Fig 3C and 3D). Down-regulation of CDC50.3 resulted in a moderate decrease in fitness (Fig 3C and 3D).

## P-type IV ATPases and its CDC50 chaperones play distinct roles in the parasite biology

To dissect the fitness conferring role of ATP7B depletion, each of the steps of the lytic cycle were examined individually. Parasites depleted in ATP7B are impaired in intracellular growth (S3A Fig) but egress and invade normally (S3B and S3C Fig). Moreover, the organization of intracellular parasites in rosettes was disrupted in the absence of ATP7B (S3D and S3E Fig). Deeper characterization would be needed to understand the importance of ATP7B in the parasite biology.

Parasites depleted in either ATP2B or CDC50.4 showed a severe impairment in invasion (Fig 4A) and in egress (Fig 4B), without alteration of intracellular growth (Fig 4C). Importantly, microneme secretion of extracellular parasites depleted in either ATP2B or CDC50.4

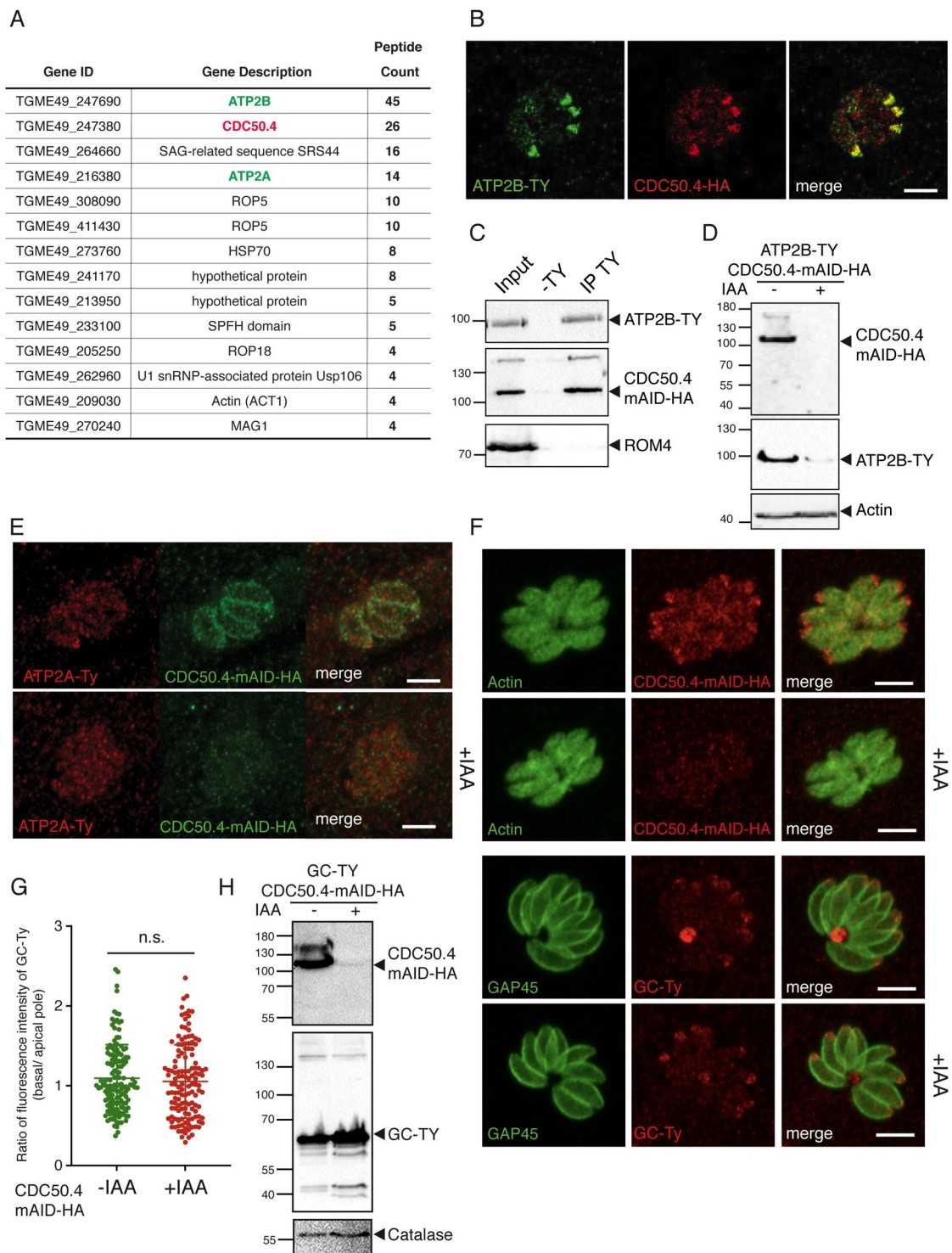

**Fig 2. ATP2B and CDC50.4 form an heterocomplex.** (A) Gene ID and number of unique peptides identified as putative interactors of CDC50.4 upon coimmunoprecipitation and mass spectrometry analysis (B) IFA of intracellular ATP2B-Ty/CDC50.4-mAID-HA parasites. (C) Western blot of immunoprecipitation with anti-Ty from ATP2B-Ty/CDC50.4-mAID-HA lysate showing that ATP2B-Ty is associated with CDC50.4-mAID-HA. (D) Western blot of lysates ATP2B-Ty/CDC50.4-mAID-HA parasites treated with or without IAA for 24 hours. Actin: loading control. (E) IFA of intracellular RH ATP2A-Ty/CDC50.4-mAID-HA parasites with or without IAA. (F) IFA of intracellular RH GC-Ty/CDC50.4-mAID-HA parasites with or without IAA. (G) Quantification of representative pictures in (F). The ratio between the intensity of fluorescence at the basal pole versus the apical pole of the parasite is shown. Approximately 100 vacuoles were quantified. (H) Western blot of lysates GC-Ty/

CDC50.4-mAID-HA parasites treated with or without IAA for 24 hours. Catalase: loading control. The scale bars for the immunofluorescence images are 7μM, unless otherwise indicated.

triggered by BIPPO (PDE inhibitor which induces the accumulation of cGMP in the cell) was considerably reduced (Fig 4D and 4E). The defect in microneme exocytosis explains the impaired egress and invasion phenotype of ATP2B, suggesting a crucial role of the heterocomplex for the completion of the parasite's lytic cycle. In contrast, depletion of the paralogue protein ATP2A did not affect any steps of the parasite lytic cycle including invasion (Fig 4F), egress (Fig 4G), intracellular growth (Fig 4H) or microneme secretion (Fig 4I and 4J).

## ATP2B-CDC50.4, but not ATP2A-CDC50.4, is a phosphatidylserine flippase at the plasma membrane

Anchoring of ATP2A-CDC50.4 and ATP2B-CDC50.4 complex to the parasite plasma membrane was demonstrated by protease protection assay on non-permeabilized parasites. Proteins exposed to the outer leaflet of the plasma membrane are susceptible to cleavage by proteases. Concomitantly, the disappearance of the full length ATP2A, ATP2B and CDC50.4 upon protease treatment demonstrates that these complexes localize to the plasma membrane of the parasite (Fig 5A–5C). The presence of the complexes at the parasite plasma membrane, offers the convenient opportunity to assess its flippase activity using a live cells assay [41] as previously reported in *T. gondii* [8]. We focused our investigation on the analysis of phosphatidylserine (PS) since we previously demonstrated that it is the phospholipid that extracellular parasites majorly incorporate into the plasma membrane [8]. A bulk time-dependent increase in nonquenchable fluorescent analogues of PS was crucially dependent on the presence of ATP2B at the plasma membrane in extracellular (Fig 5D) or intracellular mimicking conditions (S3F Fig), whereas no changes were found upon depletion of ATP2A (Fig 5E). The depletion of CDC50.4 mimicked the effects of the depletion of ATP2B with respect to the bulk PS flipping activity (Fig 5F), in contrast to CDC50.1, which did not affect bulk PS activity upon depletion (Fig 5G).

The impact of ATP2B on microneme secretion implicates the importance of a pool of PS at the inner leaflet of the plasma membrane. Such a pool can be detected using the genetically encoded molecular probe lactadherin C2 domain (Lact-C2) fused to GFP known to bind to PS [42]. GFP-Lact-C2 specifically labelled the parasite periphery (Fig 5H) as well as some endomembrane compartments where PS synthesis possibly takes place. Mutation in the specific binding site of Lac-C2 for PS inhibited the plasma membrane localization of the protein (Fig 5I) [42]. Importantly, no changes were observed in Lac-C2 localization upon depletion of ATP2B (Fig 5H). These results are not surprising since PS is the most abundant anionic phospholipid in eukaryotic membranes (accounting up to 10% of the total cellular lipids) and it is highly concentrated at the inner leaflet of the plasma membrane [43]. Concordantly, due to the high affinity of LacC2 to PS [44], low concentration of PS would be sufficient for its binding and re-localization.

We then reasoned that fluctuation in PS levels would be easier to measure in the outer leaflet of the plasma membrane, where concentration in wild type parasites is low. Compellingly, parasite depleted in ATP2B failed to restore PS asymmetric distribution in natural conditions, leading to an accumulation of PS in the outer leaflet of the plasma membrane that can be detected in extracellular parasites via binding to Annexin V (Figs 5J and S3G and S3H).

Taken together, these results demonstrate that ATP2B-CDC50.4 complex, but not ATP2A complex, controls flipping of PS at the plasma membrane of *T. gondii*.

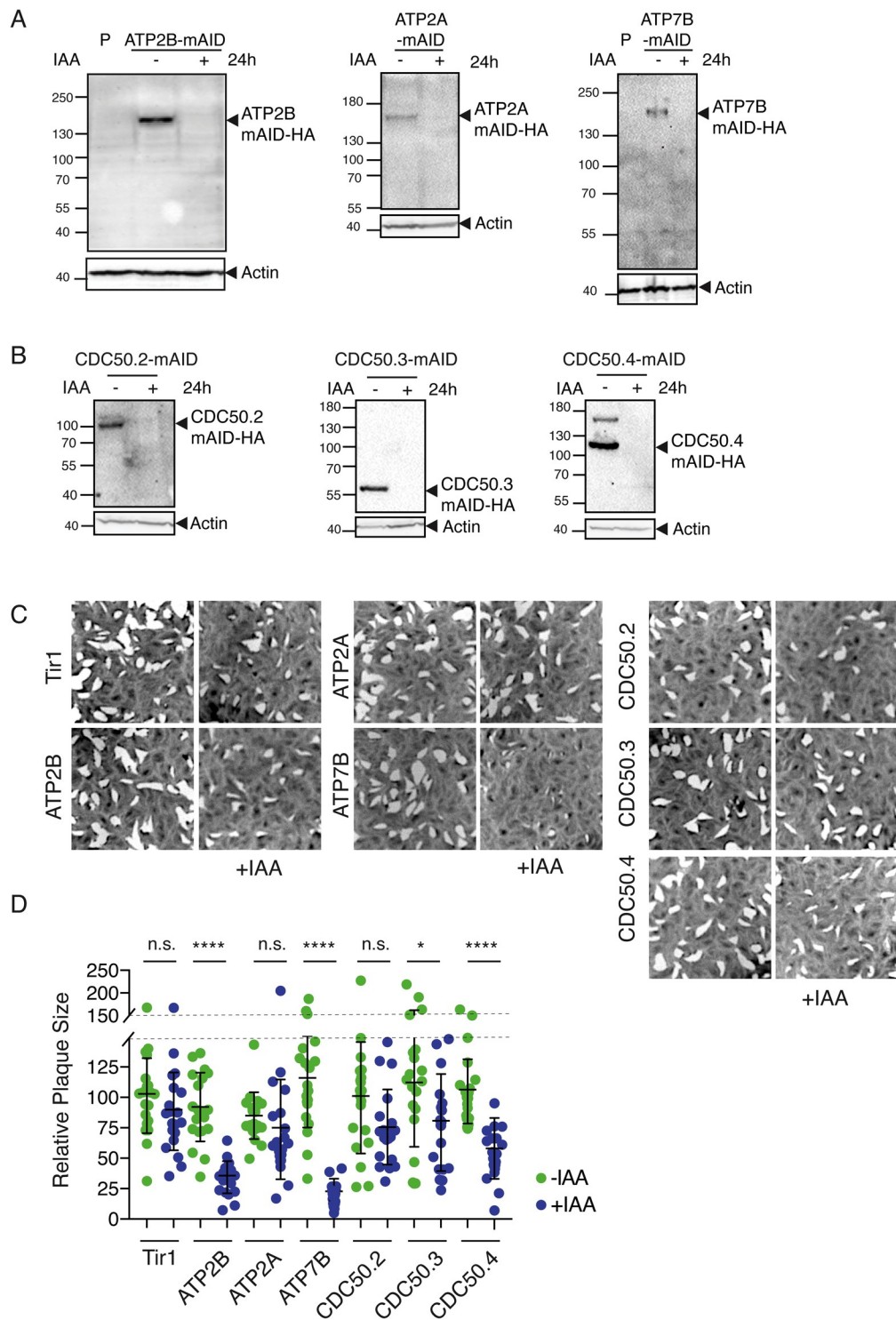

**Fig 3. Fitness conferring and dispensable P-type IV ATPases and CDC50 subunits are encoded in the *T. gondii* genome.** (A) Western blot analysis of ATP2A-mAID, ATP2B-mAID and ATP7B-mAID regulation with IAA treatment for 24 hours. Actin: loading control. P: parental strain (Tir1 strain) (B) Western blot of lysates from CDC50.2-4-mAID parasites treated with or without IAA for 24 hours. Actin: loading control. (C) Images of plaques formed by RH Tir1 parental strain, ATP2A, ATP2B, ATP7B, CDC50.2, CDC50.3 and CDC50.4-mAID-HA lines on HFF monolayers with or without IAA treatment. (D) Quantification of plaque size relative to the parental control (Tir1), mean +/- SD of 1 representative experiment. Each parasite line was analysed individually for statistical significance using an unpaired t test. P values: **** = <0.0001, * = <0.05.

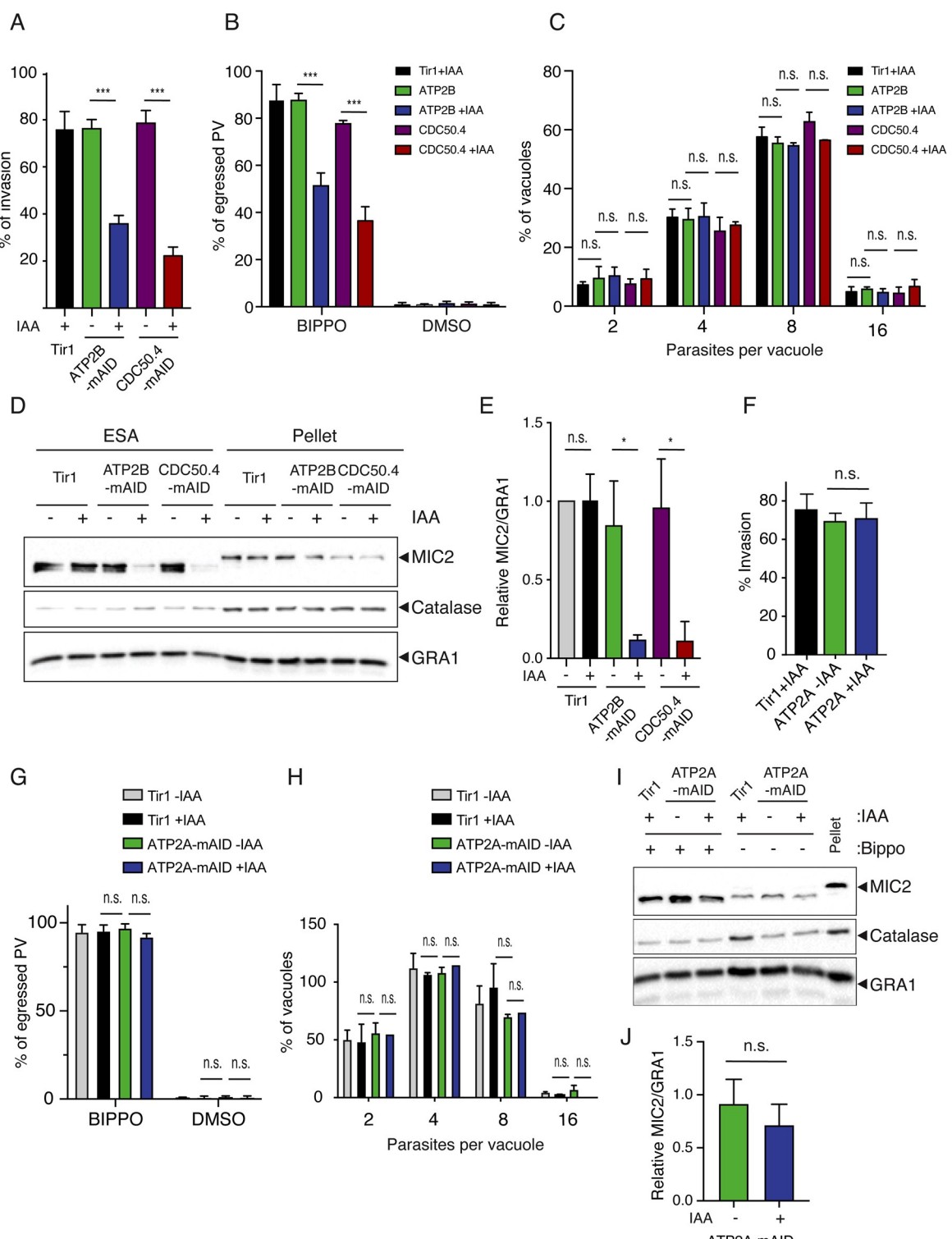

**Fig 4. ATP2B-CDC50.4 heterocomplex facilitates microneme secretion.** (A) Invasion assay of Tir1 parental strain, ATP2B-mAID-HA and CDC50.4-mAID-HA parasites treated with or without IAA for 24 hours. Data represents mean +/- SD of three independent experiments. (B) Egress assay of Tir1 parental strain, ATP2B-mAID-HA and CDC50.4-mAID-HA parasites grown for 30 hours treated with or without IAA. Egress was induced with BIPPO (PDE inhibitor which induces the accumulation of cGMP in the cell) or DMSO for 7 minutes. The percentage of egress (lysed vacuoles) is shown as means +/- SD of 3 independent replicates. (C) Parasites lacking ATP2B or CDC50.4 are not impaired in intracellular replication. Error bars represent +/- SD from three independent experiments. (D)

Microneme secretion of extracellular of Tir1 parental strain, ATP2B-mAID-HA and CDC50.4-mAID-HA parasites stimulated with or without BIPPO after having been treated or not with IAA for 24 hours. ESA (excreted-secreted antigens) and pellet fractions are shown. MIC2: microneme ESA. GRA1: dense granule ESA. Catalase: lysis control. Relative ratio of MIC2 secretion compared to Tir1 –IAA parental control +/- SD of 3 independent replicates is shown in (E). (F) Invasion assay of Tir1 parental strain and ATP2A-mAID-HA parasites treated with or without IAA for 24 hours. Data represents mean +/- SD of three independent experiments. (G) Egress assay of Tir1 parental strain and ATP2A-mAID-HA parasites grown for 30 hours treated with or without IAA. Egress was induced with BIPPO (PDE inhibitor which induces the accumulation of cGMP in the cell) or DMSO for 7 minutes. The percentage of egress (lysed vacuoles) is shown as means +/- SD of 3 independent replicates. (H) Parasites lacking ATP2A are not impaired in intracellular replication. Error bars represent +/- SD from three independent experiments. (I) Microneme secretion of extracellular of Tir1 parental strain and ATP2A-mAID-HA parasites stimulated with or without BIPPO after having been treated or not with IAA for 24 hours. ESA and pellet fractions are shown. MIC2: microneme ESA. GRA1: dense granule ESA. Catalase: lysis control. Relative ratio of MIC2 secretion compared to Tir1 –IAA parental control +/- SD of 3 independent replicates is shown in (J). Each parasite line was analysed individually for statistical significance using an unpaired t test. P values: **** = <0.0001, * = <0.05.

## Discussion

Phospholipid asymmetry plays a key role in several indispensable cellular functions including membrane potential [45], receptor based signaling [46] and secretion of vesicles [47]. P4-ATPases are central flippases that help maintain lipids asymmetry [25]. Most P type-IV ATPases usually require CDC50 partners acting as chaperones for correct localization and activity [48], and complexes formed between these proteins have been shown to be either highly promiscuous or specific [48]. Here we demonstrate that *T. gondii* encodes for 6 type IV ATPases and 4 CDC50 cofactors with different functions and fitness associated to their deletion. Importantly, some of the data presented here are supported by an overlapping study [30].

### P4-ATPases and CDC50 complex formation in Apicomplexa

Here, we demonstrate that in *T. gondii*, CDC50.4 forms heterocomplexes with ATP2B and ATP2A but not with GC despite sharing a similar localization [8], demonstrating some level of specificity in complex formation independent of protein localization. ATP2B acts as a PS flippase at the plasma membrane which plays a crucial role in microneme exocytosis. The promiscuity of ATP2B substrates was not assessed in this study and would require further analysis. Importantly, a recent report has indicated that *P. chabaudi* recombinant ATP2 is capable of flipping PS and PE [49], which would suggest that *T. gondii* ATP2B or ATP2A would also flip PE. Importantly, the authors of this study identified CDC50A and CDC50B as interactors of ATP2, while CDC50C (homologue of CDC50.4) could not be produced recombinantly [49]. This promiscuity of binding remains to be confirmed *in vivo* since we show here that *T. gondii* ATP2B is incapable of using other CDC50 chaperones to compensate for the lack of CDC50.4.

Assignment of other pairs will await further investigation but ATP7 and CDC50.3 might form another heterocomplex which is absent in *Theileria* and *Cryptosporidia*. ATP7 has previously been shown to be essential and localize to the parasite–host interface in *Plasmodium* parasites [29]. Despite not being the main focus of this study, we show here that ATP7B is important for *T. gondii* intracellular growth while ATP7A has been reported to be dispensable based on its fitness score deduced from the genome wide analysis [28]. In addition, a recent report indicates that mutations in the ATP7B are associated with increase resistance of *T. gondii* to extracellular environment during *in vitro* evolution studies [50]. The mechanistic details for this emergent resistance are obscure.

ATP8 belongs to the class 2 of P4-ATPases, which includes ATP9A and ATP9B in mammals and Neo1p in yeast [48]. These proteins do not appear to use CDC50 as β-subunit which might indicate that apicomplexan ATP8 can function independently of any CDC50 subunit. In yeast and mammals, these proteins translocate PS and affect the Golgi/endosomal system

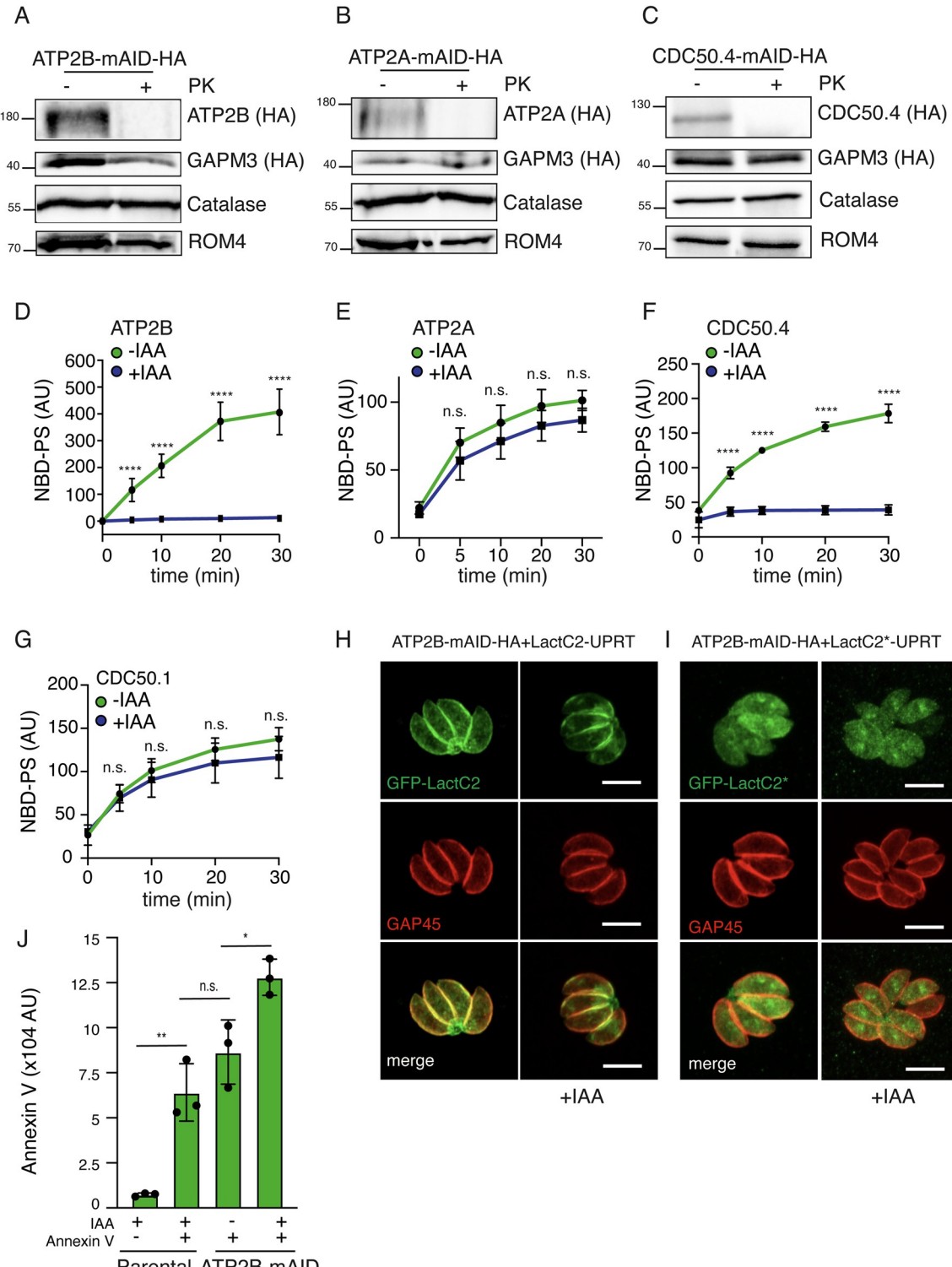

**Fig 5. ATP2A and ATP2B are phospholipid specific flippases at the parasite plasma membrane.** (A-C) Full length ATP2A, ATP2B and CDC50.4 are digested by proteinase K (PK) in non-permeabilized parasites, respectively. C-terminally HA-tagged ATP2B and CDC50.4 were used for the assay. ROM4; plasma membrane protein. GAPM3; alveolar protein. Catalase; cytosolic marker. (D-G) Flow cytometry measurement of residual fluorescence upon addition of DPX to NBD-PS incubated extracellular ATP2B, ATP2A, CDC50.4 and CDC50.1-mAID strain of *T. gondii*, respectively. Data represents mean +/- SD of three independent experiments. (H) LactC2 stains plasma membrane and internal vesicular organelles even upon downregulation of ATP2B. GAP45: parasite periphery. (I) LactC2

localized to the cytoplasm of the parasite upon mutation of PS binding sites. GAP45: parasite periphery. (J) Flow cytometry measurement of Annexin V staining of extracellular parental and ATP2B-mAID strain *T. gondii*. Data represents mean +/- SD of three independent experiments. Each parasite line was analysed individually for statistical significance using an unpaired t test. P values: **** = <0.0001, * = <0.05. The scale bars for the immunofluorescence images are 7μM, unless otherwise indicated.

[51] and recycling of endosomes [52]. A similar function of ATP8 in PS flipping at the Golgi/ endosomal compartment in apicomplexans remains to be assessed.

With the group of alveolates, the apicomplexan parasites as well as some ciliates [53] have directly fused a P4-ATPase with GC catalytic domains to form a large GC protein [5]. In *T. gondii*, CDC50.1 was shown to be essential for GC localization and the sensing and integration of external signals, notably phosphatidic acid [8]. *Plasmodium* species possess two genes that harbor a fusion with P4-ATPases GCα and GCβ. Importantly, in *P. yoelii*, CDC50A has been demonstrated to interact with GCβ and to be essential for ookinete gliding motility [54]. The *Plasmodium* CDC50A groups with the members of CDC50.3 (S1 Fig). On the other hand, the β-subunit associated to GCα remains to be identified. *Plasmodium* CDC50A and CDC50B belong to the same phylogenetic subgroup that *T. gondii* CDC50.1 and CDC50.2, and would be good candidates to bind GCα. Conflictingly, neither of these proteins are essential for *Plasmodium yoelii* erythrocytic stages (PY17X_0619700 and PY17X_0916600) [55] and, despite discrepancies with a genome wide screening on *P. falciparum* [56], would suggest a possible functional redundancy between CDC50A and CDC50B.

## Maintenance of PS asymmetry by ATP2B-CDC50.4 at the plasma membrane is crucial for efficient microneme secretion

Plasma membrane asymmetry is an essential need in cell biology [24–26] and flipping of PS is likely to be maintained across the entire lytic cycle of the parasite for survival. Coherently, ATP2B flippase activity is maintained even in intracellular mimicking conditions (S3F Fig). Here, we demonstrate that the first repercussion of the dysregulation of PS asymmetry at the plasma membrane are during egress, invasion and egress. Compellingly, PS in the inner leaflet of the plasma membrane is known to play a critical role in neurotransmitter release [47]and insulin secretion [57] in mammalian cells. Moreover, *Candida albicans* strains impaired in PS biosynthesis display decreased ability to secrete proteases and phospholipases [58]. As in most eukaryotic cells, PS is synthesized at the cytosolic leaflet of the ER in *T. gondii* [59] and asymmetry is predictably maintained by flippases at the Golgi and plasma membrane [41,60]. Golgi localized PS flippases are key players in exocytic vesicle sorting [60]. Once PS reaches the parasite plasma membrane, the ATP2B-CDC50.4 heterocomplex presumably ensures an enrichment of PS at the inner leaflet at the apical tip of the parasites. Any excess of PS at the plasma membrane is rapidly converted into PE as recently demonstrated [30]. The disruption of this homeostasis might lead to overall changes in plasma membrane tension, curvature and could also affect the activity of important signaling components (i.e. GC [8–11], PKG [14], PI-PLC or DGK1 [21]), explaining the phenotype associated to the depletion of ATP2B-CDC50.4 complex. In addition to the previously reported role of PA [21], PS is a second anionic PL implicated in the docking and/or fusion of the micronemes with the plasma membrane. PA is recognized by APH, an acylated protein at surface of the micronemes [21]. Hypothetically, a plausible candidate binding to PS could be DOC2.1 [35] or Ferlin 1 (FER1) [37], and since this phospholipid is enriched at the inner leaflet of the plasma membrane (Fig 5H), might contribute to the mechanism of recognition and fusion of micronemes with the plasma membrane for exocytosis, similarly to well-studied mechanisms proposed in model organisms [32,61,62].

In addition to its role in exocytosis, 'healthy' exposure of PS has been previously associated to pathogenesis and immune regulation by *T. gondii* [63], as well as other eukaryotic parasites [64,65]. The regulation of PS exposure at the plasma membrane of *T. gondii* and the role of ATP2B-CDC50.4 in this process remain to be investigated. Remarkably, *T. gondii* is known to secrete a soluble PS decarboxylase which might contribute to a decrease of PS concentration at the outer leaflet of the plasma membrane [66].

A possible implication of ATP2B in phosphatidylthreonine (PT) homeostasis has not been investigated due to the lack of commercially available tools to study this phospholipid. PT was previously described as highly enriched phospholipid in Apicomplexa [59] and was shown to be associated to calcium homeostasis in *T. gondii* [67]. It is possible that the phenotype of ATP2B is, at least partially, associated to an unexplored capacity of ATP2B to translocate PT. On the other hand, PT synthesis was previously shown to impact specifically natural egress [59], while induced egress reminds unaltered [67]. In addition, since the lack of PT could not be complemented nor aggravated by excess or reduction of PS [59], it is not likely that PS and PT have redundant functions for secretion. Taken together, this data strongly indicates that lack of PS translocation is the main responsible of the phenotype associated to the depletion of ATP2B-CDC50.4 complex showed here.

Overall, this study identified the complex ATP2B-CDC50.4, which is a PS flippase that crucially contributes to motility, invasion and egress. A model by which PS concentration at the inner leaflet of the plasma membrane contribute to microneme docking and exocytosis could imply the participation of lipid binding proteins. However, this hypothesis awaits further investigations.

## Materials and methods

### Bacteria, parasite and host cell culture

E. coli XL-10 Gold chemo-competent bacteria were used for all recombinant DNA experiments. Parental *T. gondii* strain Ku80 KO (genotype RHΔhxgprtΔku80) and parental parasites expressing the Tir1 protein were used in this study [14]. *T. gondii* tachyzoites parental and derivative strains were grown in confluent human foreskin fibroblasts (HFFs) maintained in Dulbecco's Modified Eagle's Medium (DMEM, Gibco) supplemented with 5% fetal calf serum (FCS), 2 mM glutamine and 25 mg/ml gentamicin. Depletion of mAID fusion proteins was achieved with 500 μM of IAA [14].

### Preparation of T. gondii genomic DNA

Genomic DNA (gDNA) was prepared from tachyzoites of RH or RH ΔKu80 (here referred as Δku80) strains using the Wizard SV genomic DNA purification (Promega) according to manufacturer's instructions.

### DNA vector constructs and transfection

All primers used in this study are listed in S2 Table. Auxin-inducible degradation of ATP2A, ATP2B, ATP7B, CDC50.2, CDC50.3 and CDC50.4 were generated using a PCR fragment encoding the mAID–HA and the HXGPRT cassette produced using the KOD DNA polymerase (Novagen, Merck) with the vector pTUB1:YFP-mAID-3HA as template and the primers indicated in S2 Table. A specific sgRNA was generated to introduce a double-stranded break at the 3′ of each gene (primers used to generate the guide are indicated in S2 Table).

Parasite transfection and selection of clonal stable lines *T. gondii* tachyzoites were transfected by electroporation as previously described [68]. Selection of transgenic parasites were

performed either with mycophenolic acid and xanthine for HXGPRT selection [69], pyrimeth-amine for DHFR selection [70] or chloramphenicol for CAT selection [71]. Stable line for all expressing strains were cloned by limited dilution and checked for genomic integration by PCR and analysed by IFA and/or WB.

## Antibodies

The monoclonal antibodies against the Ty tag BB2 (1:10 dilution by WB, 1:20 by IFA) [72], actin (1:20 WB) [72], SAG1 (1:20 IFA) (generous gift from J-F. Dubremetz), GRA1 T5-2B4 (1:50 WB, 1:100 IFA), GRA3 (1:50 WB, 1:100 IFA) (generous gift from J-F. Dubremetz), MIC2 (1:20 WB, 1:50 IFA) (generous gift from J-F. Dubremetz), MIC3 T4-2F3 (1:20 WB, 1:50 IFA), anti-Catalase (1:2000 WB) [73], anti-IMC1 (1:2000 WB, 1:1000 IFA), anti-ARO (1:3000 IFA). For western blot analysis, secondary peroxidase-conjugated goat anti-rabbit-IgG, anti-mouse-IgG antibodies and secondary Alexa-Fluor-680-conjugated goat anti-rabbit IgG antibodies (Thermofisher) were used. For immunofluorescence analysis, the secondary antibodies Alexa-Fluor-488-conjugated goat anti-rabbit IgG antibodies, Alexa-Fluor-488-conjugated goat anti-mouse-IgG antibodies and Alexa-Fluor-594-conjugated goat anti-mouse-IgG antibodies (Thermofisher) were used.

## Immunofluorescence assay (IFA)

HFFs seeded on coverslips in 24-well plates were inoculated with freshly egressed parasites. After 24 h, cells were fixed with 4% paraformaldehyde (PFA) and 0.005% glutaraldehyde (GA) in PBS for 10 min and processed as previously described [8]. Confocal images were acquired with a Zeiss confocal laser scanning microscope (LSM700 or LSM800) using a Plan-Apochromat 63x objective with NA 1.4 at the Bioimaging core facility of the Faculty of Medicine, University of Geneva. Final image analysis and processing was done with Fiji [74].

## Western blotting

Freshly egressed parasites were pelleted after complete host cell lysis. SDS-PAGE, wet transfer to nitrocellulose and proteins visualized using ECL system (Amersham Corp) were performed as described previously [8].

## Plaque assay

A confluent monolayer of HFFs was infected with around 50 freshly egressed parasites for 7 to 8 days before cells were fixed with PFA/GA. Plaques were visualized by staining with Crystal Violet (0.1%) as previously described [8]. Quantification was performed using the Fiji [75].

## Intracellular growth assay

Parasites were allowed to grow on HFFs for 24 h prior to fixation with PFA/GA. IFA was performed as described previously [8].

## Invasion assay

Freshly egressed parasites were inoculated on coverslips seeded with HFFs monolayers and centrifuged at 1100 x g for 1 min. Invasion was allowed for 20 min at 37˚C +/- ATc prior to fixation with PFA/GA. Extracellular parasites were stained first using monoclonal anti-SAG1 Ab in non-permeabilized conditions. After 3 washes with PBS, cells were fixed with 1% formaldehyde/PBS for 7 min and washed once with PBS. This was followed by permeabilization with 0.2% Triton/PBS and staining of all parasites with polyclonal anti-GAP45 Ab. Appropriate

secondary Abs were used as previously described. 100 parasites were counted for each experiment, the ratio between red (all) and green (invaded) parasites is presented. Results are presented as mean ± standard deviation (SD) of three independent biological replicate experiments.

### Induced egress assay

Freshly egressed tachyzoites were added to a new monolayer of HFFs, washed after 30 min and grown for 30 h. The infected HFFs were washed once in serum-free DMEM and then incubated with 50 μM BIPPO in serum-free DMEM for 7 min at 37˚C. Cells were fixed with PFA/GA and processed for IFA using anti-GAP45 Ab. 100 vacuoles were counted per strain and scored as egressed or non-egressed. Results are presented as mean ± standard deviation (SD) of three independent biological replicate experiments. Control experiment with DMSO showed no egress. For live video microscopy of induced egress, parasites were grown on glass bottom plates seeded with HFFs monolayers for 30 h at 37˚C and egress was induced as described above.

### Microneme secretion assay

Microneme secretion assay was performed on freshly egressed parasites, pre-treated 24 or 48 h +/- ATc. Parasites were pelleted at 1000 rpm for 5 min and resuspended in extracellular (EC) buffer (142 mM NaCl, 5.8 mM KCl, 1 mM $MgCl_2$, 1 mM $CaCl_2$, 5.6 mM glucose, 25 mM HEPES, pH to 7.2 with NaOH). After centrifugation, the pellets were resuspended in 100 μL of extracellular (EC) buffer containing +/- 2% ethanol and incubated for 30 min at 37˚C. Then, parasites were pelleted at 1000 x g for 10 min at 4˚C, the supernatant was transferred to a new Eppendorf tube (the pellet from this step serves as the pellet fraction) and centrifuged again at 2000 x g for 10 min at 4˚C. The final supernatant, containing the excreted/secreted antigens (ESA), and pellet fraction were resuspended in SDS loading buffer and boiled prior to immunoblotting.

### Flippase assay

NBD-phospholipid incorporation (NBD-PS) was assessed by flow cytometry as described before [8,41]. In brief, $5 \times 10^6$ extracellular parasites were washed in Hank's balanced salt solution (pH 7.4) containing 1 g l–1 glucose. Subsequently, 1 μM NBD-PS was incubated at room temperature. At the designated time point, 20 mM DPX (p-xylene-bis-pyridinium bromide) was added to quench fluorescence of lipids localized in the outer leaflet. Then, 10,000 cells were analysed with a Gallios (4-laser) cytometer. The mean fluorescence intensities of the cells were calculated.

### Immunoprecipitation assay

Extracellular tachyzoites were harvested, washed in PBS and lysed in co-immunoprecipitation buffer (0.2% v/v Triton X-100, 50 mM Tris-HCl, pH 8, 150 mM NaCl) in the presence of a protease inhibitor cocktail (Roche). Cells were sonicated on ice and centrifuged at 14,000 r.p.m. for 30 min at 4˚C. Supernatants were then subjected to immunoprecipitation using anti-HA antibodies as previously described [8]. 2 μl of DTT (50 mM in liquid chromatography–mass spectrometry-grade water) were added and the reduction was carried out at 37˚C for 1 h. Alkylation was performed by adding 2 μl of iodoacetamide (400 mM in distilled water) for 1 h at room temperature in the dark. Protein digestion was performed overnight at 37˚C with 15 μl of freshly prepared trypsin (Promega; 0.2 μg μl–1 in ammonium bicarbonate). After beads

were removed, the sample was desalted with a C18 microspin column (Harvard Apparatus), dried under speed vacuum, and redissolved in H2O (94.9%), CH3CN (5%) and FA (0.1%) before liquid chromatography–electrospray ionization-tandem mass spectrometry analysis (LC–ESI-MS/MS). LC–ESI-MS/MS was performed on a Q-Exactive Hybrid Quadrupole-Orbitrap mass spectrometer (Thermo Fisher Scientific) equipped with an Easy nLC 1000 system (Thermo Fisher Scientific). Peptides were trapped on an Acclaim PepMap 100, C18, 3 μm, 75 μm × 20 mm nano-trap column (Thermo Fisher Scientific) and separated on a 75 μm × 500 mm, C18, 2 μm Easy-Spray column (Thermo Fisher Scientific).

### Annexin V staining

Annexin V (ThermoFisher, 88-8005-72) labelling was performed as indicated by supplier. Briefly, $1x10^6$ freshly egressed parasites were resuspended in binding buffer provided by supplier. 5 uL of Annexin V was added for labelling and incubated during 10–15 minutes. Upon washing once, 10,000 cells were analyzed with a Gallios (4-laser) cytometer. The mean fluorescence intensities of the cells were calculated.

### In silico analysis of proteins and modelling

Sequences of Apicomplexan P-type ATPases and CDC50s were procured from EuPathDB and aligned using MUSCLE sequence alignment software [76,77]. The resulting sequence alignment was manually curated utilizing BioEdit (http://www.mbio.ncsu.edu/bioedit/bioedit.html) to edit out uninformative alignment positions. Phylogeny tree was generating utilizing PhyML [78] on the curated MUSCLE alignment, using LG model of amino acids substitution with NNI topology search. Phylogeny.fr [78] platform was utilized for much of the above analysis. All accession numbers are provided in S1 Table.

C2 domain modelling of TgDOC2.1 was performed using the automated server i-TASSER [79] and visualized using PyMOL (www.pymol.org). Modelling was performed using the residues 569 to 653, using the sequences 4ihbA, 3jzyA, 4icw, 5ixcA, 4rj9A, 3pfqA and 5ixcA (Swissmodel templates) with normalized Z-scores of 0.87 to 2.51. Overall model possesses an estimated C-score of -0.41, TM score of 0.66 ± 0.13 and RMSD of 4.4 ± 2.9Å.

### Statistics and reproducibility

All data are presented as the mean ± s.d. of 3 independent biological replicates (n = 3), unless otherwise stated in the figure. The mean of each independent biological replicate was generated by counting 100 vacuoles/parasites. All data analyses were carried out using GraphPad Prism. The null hypothesis (α = 0.05) was tested using unpaired two-tailed Student's t-tests and significant P values are shown.

## Supporting information

**S1 Fig.** (A) Apicomplexan CDC50s cluster into 2 phylogenetic groups. An unrooted maximum likelihood tree of apicomplexan ASPs was generated using PhyML v3.0, using WAG model of amino acids substitution with NNI topology search, based on an amino acid alignment by MUSCLE. The genes are represented by the EuPathDB accession numbers. Node support values are indicated.
(TIF)

**S2 Fig.** (A) PCR demonstrates correct integration of ATP2A, ATP2B and ATP7B and CDC50.2–4 mAID. Primers used are listed in S2 Table. (B) PCR demonstrates correct integration of ATP7B-SM-HA. Primers used are listed in S2 Table. (C) Immunoblot of lysates from

RH parental and ATP7B-SM-HA parasites. HA antibodies were used to detect tagged ATP7B. MIC2: loading control. (D) Western blot showing enrichment of CDC50.4-mAID-HA upon immunoprecipitation. Anti-HA antibodies were used to detect CDC50.4-mAID-HA in the different fractions. S: soluble fraction (input), P: pellet fraction. (E) IFA of intracellular RH ATP2B-Ty/CDC50.4-mAID-HA parasites with or without IAA. The scale bars for the immunofluorescence images are 7μM, unless otherwise indicated. (F) Western blot of lysates ATP2A-Ty/CDC50.4-mAID-HA parasites treated with or without IAA for 24 hours. Catalase: loading control.
(TIF)

**S3 Fig.** (A) Parasites lacking ATP7B display a delay in intracellular replication. Error bars represent ±SD from three independent experiments. (B) Egress assay of Tir1 parental strain and ATP7B-mAID-HA parasites grown for 30hs treated with or without IAA. Egress was induced with BIPPO or DMSO for 7 minutes. Percentage of egressed vacuoles is shown as means+/- SD of 3 independent replicates. (C) Invasion assay of Tir1 parental strain and ATP7B-mAID-HA parasites treated with or without IAA for 24 hours. Data represents mean +/- SD. (D) Representative images of vacuole organization in parasites depleted (or not) of ATP7B. Quantification is shown in (E). (F) Flow cytometry measurement of residual fluorescence upon addition of DPX to NBD-PS incubated extracellular ATP2B-mAID in intracellular buffer. Parasites were mechanically released from intracellular condition to avoid activation of egress signalling. (G-H) Histograms corresponding to one experiment of Annexin V binding to parasites are shown in (G) and representative images in (H). The scale bars for the immunofluorescence images are 7μM, unless otherwise indicated.
(TIF)

**S1 Table. Gene accession numbers of the homologs of the studied genes within the Apicomplexa phylum.**
(XLSX)

**S2 Table. Oligonucleotide sequences used in this study.**
(XLSX)

## Author Contributions

**Conceptualization:** Hugo Bisio, Dominique Soldati-Favre.

**Data curation:** Hugo Bisio, Aarti Krishnan, Jean-Baptiste Marq.

**Formal analysis:** Aarti Krishnan.

**Funding acquisition:** Dominique Soldati-Favre.

**Investigation:** Hugo Bisio, Aarti Krishnan, Jean-Baptiste Marq.

**Methodology:** Hugo Bisio.

**Project administration:** Dominique Soldati-Favre.

**Validation:** Hugo Bisio, Aarti Krishnan, Jean-Baptiste Marq.

**Visualization:** Hugo Bisio, Aarti Krishnan.

**Writing – original draft:** Hugo Bisio, Dominique Soldati-Favre.

**Writing – review & editing:** Aarti Krishnan.

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
