## [Decision Letter · Decision Letter 0]

23 May 2021

Dear Dominique,

Thank you very much for submitting your manuscript "Toxoplasma gondii phosphatidylserine flippase complex ATP2B-CDC50.4 critically participates in microneme exocytosis" for consideration at PLOS Pathogens. As with all papers reviewed by the journal, your manuscript was reviewed by members of the editorial board and by several independent reviewers. In light of the reviews (below this email), we would like to invite the resubmission of a significantly-revised version that takes into account the reviewers' comments.

As you will see, all three reviewers are excited by the study and agree that it makes an important contribution to understanding of the role of P4-ATPase flippases in phospholipid asymmetry, microneme discharge and host cell invasion by Toxoplasma. However, the reviewers also share a number of substantial concerns with the manuscript in its current form. In particular, they agree that not all the conclusions are currently supported by the experimental data. Areas in which additional experimental evidence is requested include the suggestion that PS binds directly and specifically to DOC2.1. Examining the effects of ATP2B knockdown on DOC2.1 localisation would also inform the major conclusions, as would analysis of the effects of CDC50.4 knockdown on subcellular localisation of ATP2A and ATP2B. An examination of whether PS flipping occurs in intracellular parasites would also address the proposed link with egress. Whether these P4-ATPases can flip phospholipids other than PS has not been examined, although we do not feel this is an essential requirement. More detailed analysis of the calcium dependency of DOC2.1-PS binding is warranted. The inclusion of statistical analysis of the quantitative data (including full details in the Methods section of the statistical methods used), as well as information on numbers of experimental replicates, is essential. Finally, Figure 6 in particular requires attention and corrections, whilst a number of textual changes and corrections are requested throughout the manuscript that we agree in most cases would help to clarify the narrative.

We cannot make any decision about publication until we have seen the revised manuscript and your response to the reviewers' comments. Your revised manuscript is also likely to be sent to reviewers for further evaluation.

Sincerely,

Michael J Blackman

Associate Editor

PLOS Pathogens

Vern Carruthers

Section Editor

PLOS Pathogens

Kasturi Haldar

Editor-in-Chief

PLOS Pathogens

orcid.org/0000-0001-5065-158X

Michael Malim

Editor-in-Chief

PLOS Pathogens

orcid.org/0000-0002-7699-2064

As you will see, all three reviewers are excited by the study and agree that it makes an important contribution to understanding of the role of P4-ATPase flippases in phospholipid asymmetry, microneme discharge and host cell invasion by Toxoplasma. However, the reviewers also share a number of substantial concerns with the manuscript in its current form. In particular, they agree that not all the conclusions are currently supported by the experimental data. Areas in which additional experimental evidence is requested include the suggestion that PS binds directly and specifically to DOC2.1. Examining the effects of ATP2B knockdown on DOC2.1 localisation would also inform the major conclusions, as would analysis of the effects of CDC50.4 knockdown on subcellular localisation of ATP2A and ATP2B. An examination of whether PS flipping occurs in intracellular parasites would also address the proposed link with egress. Whether these P4-ATPases can flip phospholipids other than PS has not been examined, although we do not feel this is an essential requirement. More detailed analysis of the calcium dependency of DOC2.1-PS binding is warranted. The inclusion of statistical analysis of the quantitative data (including full details in the Methods section of the statistical methods used), as well as information on numbers of experimental replicates, is essential. Finally, Figure 6 in particular requires attention and corrections, whilst a number of textual changes and corrections are requested throughout the manuscript that we agree in most cases would help to clarify the narrative.

Reviewer's Responses to Questions

**Part I - Summary**

Reviewer #1: The manuscript 'Toxoplasma gondii phosphatidylserine flippase complex ATP2B-CDC50.4 critically participates in microneme exocytosis' presents some nice cell biological work utilising reverse genetics to detail the role of flippases in the asexual cycle of Toxoplasma. The authors identify P4-ATPase flippase proteins and their partner CDC50 chaperones and identify a potential, novel and essential role for an ATP2B-CDC50.4 complex in microneme secretion required for host cell invasion. Overall, the work present is well designed and the conclusions are well supported by the data presented.

Reviewer #2: The work by Bisio and colleagues provides important insights into P4-ATPases in Toxoplasma gondii - a class of enzyme that is involved in flipping phospholipids across membrane bilayers. Given previous findings that have shown that phosphatidic acid is important for mediating egress, this piece of work looks into the involvement of several P4-ATPases and their CDC50 chaperones, focusing mainly on egress and invasion. Epitope tagging and immunoprecipitation revealed that two ATPases, ATP2A and ATP2B, interact with the chaperone CDC50.4. Using the conditional auxin degradation system, Bisio et al reveal that CDC50.4 is important for ATP2B stability, but does not affect another P4-Type ATPases, GC. Using this system the authors go on to reveal that conditional knockdown of CDC50.4 and ATP2B leads to egress and invasion defects as well as a defect in microneme secretion. On the other hand, ATP7B knockdown leads to a disruption of intracellular growth and vacuole organisation, while disruption of ATP2A shows no defect at all.

Further characterisation of ATP2B and CDC50.4 revealed that ATP2B is involved in flipping phosphatidyl serine from the outer to the inner leaflet of the plasma membrane and that disruption of ATP2B leads to an accumulation of PS on the surface of tachyzoites. Next the authors take a side step and investigate the role of Doc2.1, a C2 containing protein involved in mediating microneme secretion, in sensing PS. The authors show that this protein does indeed bind to PS in a calcium independent manner. Overall these findings are important and show that regulating PS asymmetry is important in regulating processes such as egress and invasion and will be of great value to the Toxoplasma field. Despite these important findings, the manuscript in its current state requires significant improvement and amendments for publication. Most importantly: more experiments are required for ATP2A and CDC50.4 given that these are the major focus of this study, while the Doc2.1 data feels a bit like an add-on.

Reviewer #3: This work studies the role of a Toxoplasma gondii complex formed between the predicted flippase ATP2B with the cell division control protein (CDC) 50.4 for the flipping and asymmetric distribution of phosphatidylserine, which may act as lipid mediator for organelle fusion and microneme exocytosis. Authors first identified and localized putative flippases and CDC50s in T. gondii. They did this by C-terminal tagging of the respective proteins. A number of them localized to the apical end of the parasite. They decided to focus on CDC50.4 because of its predicted essentiality and conservation across the apicomplexans. They first demonstrated the interaction of CDC50.4 with ATP4A and ATP4B by immunoprecipitation coupled with mass spectrometry analysis. They tagged ATP2B in the CDC50.4-AID-HA background for co-localization studies. To study the function of these proteins they tagged them with the auxin-inducible degron (mAID) for downregulation. Downregulation of CDC50.4 led to degradation of ATP2B as shown by western blot analysis. They did not check for ATP4A. They next determined that the complex between ATP2B-CDC50.4 and also ATP7B were critical for T. gondii growth. They examined every step of the lytic cycle: invasion, egress and replication. Depletion of ATP2B or CDC50.4 caused impairment in invasion and egress but did not alter intracellular replication. Microneme secretion of extracellular parasites was reduced upon downregulation of either ATP2B or CDC50.4. No motility assays were shown. They showed that both proteins localize to the parasite membrane using a protease protection assay in non-permeabilized parasites. They measured flippase activity in situ with PS analogues, which was dependent on the presence of ATP2B at the PM. They also used a genetically encoded lactadherin C2 domain fused to GFP, known to bind PS. No controls for specific labeling of PS were shown. In conclusion authors claim to have identified a PS flippase that contributes to motility, invasion and egress. The authors claim that DOC2.1, a previously described key egress and invasion factor, senses changes in cytosolic calcium in intracellular parasites and may be the plausible sensor of PS at the inner leaflet of the plasma membrane.

The work has some weaknesses concerning statistical analysis of the data and only partial information on the number of biological replicates for each experiment. Some statements are not supported by the data presented like the Ca2+ sensing of DOC2.1 or the motility defects of the mutants. Some improvements are suggested, and more explanations and discussion of results obtained are needed.

**Part II – Major Issues: Key Experiments Required for Acceptance**

Reviewer #1: The work characterising the essentiality of identified CDC50 and P4-ATPase proteins has been carried out well. This has been recently published in an overlapping study (Ref 30) and therefore there should be further discussion of this published work included in the manuscript. However, the present study adds a detailed view of the role of these proteins at specific stages of the asexual cycle. The authors clearly demonstrate that ATP2B and CDC50.4 forms an essential complex and in some very nice experimental work, show that this complex translocates phosphatidylserine (PS). They then go on to mechanistically characterise the role of the ATP2B-CDC50.4 complex and PS translocation. The authors indicate that PS translocation by ATP2B-CDC50.4 is required for microneme secretion, and this is carried out by DOC2.1 binding of PS. Whilst this is a nice conclusion, the data presented to support the premise of DOC2.1-PS binding is weak and requires more substantial affirmation. Given that liposome-binding experiments were performed with parasite lysate, it cannot be discounted that DOC2.1 binding is not direct but mediated by another protein factor, the text should be amended to reflect this.

Additionally, since PS is an anionic lipid, it is possible that liposome binding of DOC2.1 from lysate is simply due to charge-charge interaction. The group’s previous work has demonstrated the production of the anionic lipid phosphatidic acid (PA) at the plasma membrane preceding egress, which is required for microneme secretion. Given this, the authors should repeat the sedimentation experiment of tagged DOC2.1 using liposomes containing PA to rule out that interaction is not PS specific but due to binding of anionic charge.

Furthermore, it would add more support to the hypothesis of DOC2.1-PS interaction if the study could visualise the localisation of tagged DOC2.1 in egressed parasites with and without knock-down of ATP2B. Presumably, the localisation of DOC2.1 would become increased in plasma membrane proximity which would likely be lacking upon ATP2B knock-down.

The authors outline in Figure 5A and 5B that invasion and egress are impaired in ATP2B and CDC50.4 knock-down parasites. However, looking at the data it seems that invasion is similarly impaired to the extent of the lack of egress. Can the authors rule out that invasion occurs normally, and the reduction seen is merely a result of the reduction in ATP2B and CDC50.4 knock-down in parasite egress?

The authors suggest that there is no significant effect on microneme secretion by ATP2A knock-down, however, in Figure 5I there appears to be a ~50% reduction in MIC2 secretion in ATP2A knock-down parasites in the presence of BIPPO. Could the authors comment on this? Is this reproducible? If so, this needs to be acknowledged in the main text.

Figure 6H is hard to interpret. It is unclear what the significance value relates to. Additionally, it is difficult to clearly see the increase in annexin V binding in ATP2B knock-down tachyzoites. Perhaps displaying the data as a box plot would increase clarity.

Figure 6I shows the potential localisation of PS in WT parasites, however was the visualisation of internal PS attempted in ATP2B or CDC50.4 knock-down parasites? If so, was there any difference? If not, a comment should be included in the manuscript.

Lines 87-89 Although there are references cited for the statement regarding phosphodiesterases (PDEs) : ‘…presumably regulated by PKA-C1 and cAMP…’, it leaves the reader wondering why the authors presume this. The statement either should be omitted or modified to outline briefly any evidence that regulation of TgPDEs may be by cAMP/PKA.

Line 186, this results section would benefit from brief mention of some of the other proteins identified in the pull down/MS experiment shown in Figure 2.

Line 220, should explain that proteins exposed at the surface of the PM are susceptible to the protease (rather than this just indicating PM localisation).

Line 296-298, it should be clarified that this statement refers to P. yoelii erythrocytic stages (Ref 52) rather than ‘Plasmodium species’ in general.

The section describing the results shown in Figure 5 would benefit from briefly mentioning and explaining the use of the PDE inhibitor BIPPO. I note that its use is mentioned but not explained in the figure legend.

Figure 7, what is the likely explanation for the fact that mutation of the two aspartates in DOC2.1 causes reduced fitness but not invasion or microneme secretion? Could it e.g. mean that these residues (and perhaps calcium binding) have a role in parasite growth, but not at the stage of invasion/microneme secretion?

Line 277, were different concentrations of ‘calcium’ tested to decide on the use of 0.1 mM? If not, can it be concluded from this ~30% reduction in bound protein that binding is independent of calcium?

Reviewer #2: Major Issues:

• Line 173: in this section, the authors focus on ATP2B but not ATP2A and it is unclear why the experiments were done solely on ATP2B, despite evidence of ATP2A also binding to CDC50.4. It would be interesting to see if ATP2A mislocalises when CDC50.4 is knocked down or whether there are other chaperones that can compensate for loss of CDC50.4.

• Line 183: the authors demonstrate that knockdown of CDC50.4 leads to a knockdown of ATP2B levels but they do not show an IFA of localisation of ATP2B-Ty following treatment with IAA. Does the remaining protein localise to the apical end of the parasite or is the localisation affected?

• Line 189: the authors offer no explanation of the lines that were generated. Please add a sentence explaining to the readers that the proteins of interest were fused to a mAID tag

• Line 208 & Figure 5D: the authors state in Line 208 that they assessed microneme secretion by stimulating with ethanol, however the figure legend of Figure 5D states that the parasites were stimulated with BIPPO. Can the authors correct either the legend or the main text since it is unclear which stimulant was used.

• Line 223: The authors decide to focus solely on PS experiments on ATP2A and ATP2B. Given their proven ability to do these assays with other phospholipids (PA and PC in the Bisio et al 2019 paper) and also given that they later state in the discussion that these ATPases may be involved in flipping PE, the authors should do these flipping experiments with PA, PC and PE as these would be highly informative and reduce the some speculation in the discussion section. Despite ATP7B not being involved in egress or invasion, it would also be useful for the greater Toxoplasmosis field and our understanding of Toxoplasma biology to know whether this ATPase is flipping any phospholipids, so it would also be important to include these experiments too since these lines are available.

• Line 227: the authors look at PS flipping in extracellular parasites. It would also be important to look at whether this flipping occurs in intracellular parasites i.e. by performing this experiment on parasites lysed in ENDO buffer. This is an important experiment to do to support the author’s model of PS accumulating on the inner leaflet prior to egress, since if this flipping is also occurring in intracellular parasites without any stimulation then in theory there shouldn’t be any PS flipping onto the inner leaflet.

• Line 234: the authors provide little commentary on the results from the Lact-C2 experiment - is there a difference in binding between the ATP2BmAID mutants +/-IAA? It also appears they have mixed up the Figures since Figure 6H is the annexin experiment mentioned further down and figure 6I is the Lact-C2 experiment. Furthermore in Figure 6I it is unclear which parasites are shown in the image and there should be images comparing ATP2B parasite - and + IAA. Not to mention that this subfigure (6I) has no figure legend.

• Line 235: the authors state that ATP2B knockdown parasites are unable PS asymmetric distribution - are there images to support this? The graph shown (Figure 6H shows that there is maybe more binding of Annexin V) but it would aid the reader if images of these parasites are shown. Is this increased accumulation of Annexin at the apical tip where ATP2B is localised to? Please provide these images as this is critical information that the reader should be able to see.

• Line 273: have the sedimentation experiments been done with the mutant versions of Doc2.1? If not the authors should perform these experiments since there appears to be a difference in the binding of PS + and - Ca2+ (despite it not being the pattern that the authors expected).

• Line 298: the authors reference the Jian et al paper from 2020 as the source for CDC50A and CDC50B not being essential for Plasmodium - however these experiments are nowhere to be found in this source. The authors should reference the correct source here. As far as I am aware there is only evidence of CDC50A not being essential in P. yoelii (Gao et al 2018) but I couldn’t find one for CDC50B

• Line 298: after checking PlasmoDB and looking at the mutagenesis screen by John Adams lab, it appears that CDC50B is essential and CDC50C is important for asexual growth. The authors state here that CDC50B is not essential and do not mention CDC50C in the discussion

• Line 353: the authors mention the recent FER1 pre-print, but this text seems out of place and needs more explanation or to be rewritten to link it to the rest of the discussion

• The discussion needs substantial improvements. At times it is difficult to follow the order and reasoning behind the information included. The authors should re-write this and also include summaries of their experiments. For example there is no mention of ATP7B or any of the Doc2.1 experiments which constitute a small yet significant part of the paper.

• Line 595: the authors should change "(D-F)" to (D-G) since figure G is also a part of this series of experiments.

• Line 597: the authors state in the legend that figure 6G represents the staining oft he plasma membrane and internal organs of the parasite with LactC2 but the actual figure doesn’t correspond to this. Can the authors please add the appropriate image and fix the legend.

• Line 599: in Figure 6 there is a subfigure 6I, however there is no reference to this subfigure in the figure legend. Please add a description of this image explaining what the parasite line is.

• Figure 5: no statistical analysis for any of the graphs after figure 4 are provided. The authors should provide this and if there is a statistically significant difference in the levels of PS binding of Doc2.1 + and - Ca2+ (which it looks like there is), then this should be commented on in the manuscript.

• Throughout the manuscript no statistical tests were performed. I would expect the appropriate stats tests to be performed for all graphs without exception. Without this data, the reader is missing crucial bits of information to make their own judgements on the data.

• Methods section: there is no methods section for the AnnexinV binding experiments - please add this.

Reviewer #3: Major issues

1. Figure 2E shows a lot of background with the GC-TY. Any explanation for that?

2. The IFA image shown for the downregulation of CDC50.4 appears to show some effect on the expression/distribution of GC and it concentrates in the residual body?. This result needs quantification and a better description in the legend.

3. There are no attempts to localize ATP2B in the CDC50.4 knock downs.

4. Figure 3C: the plaques for CDC50s are strange. Either there are no plaques in the image shown or they are not fully lysed. It will be good to explain what was measured and quantified for part D. Indicate in the legend the number of biological replicates.

5. Figure 4 presents the phenotype of ATP7B and is not clear how it fits in the story. Authors do not give an explanation of the disrupted rosette phenotype observed.

6. Figure 5, please indicate in the legend the number of biological replicates for all panels (some are missing) and show statistical significance analysis for the ones that are relevant.

7. Figure 6: The ATP2B-CDC50.4 complex localizes to the apical end or the plasma membrane? This should be consistent in the description of the results. The proteolysis experiment shows that the HA epitope is degraded but not necessarily the whole proteins.

8. Figure 6D-F: indicate the number of biological replicates and any statistical significance obtained in the legend. Labeling of panels need to be corrected in the figure which does not agree with the legend.

9. Figure 6H: The flow cytometry charts could be shown as supplemental. The differences in the violin plots results are not clearly evident from the distribution. Need to detail in the legend, the number of independent replicates and explain more how the statistical analysis was done. The p value shown is for which comparison?

10. Figure 6H and I are swapped in the text.

11. Figure 6I: Some evidence of specific binding is needed. Is there a cell line depleted of PS that could serve as control for the binding of the lactadherin-GFP? Reference 59 generated mutants for PS synthesis although the PS content of the mutants was not clear, and it was a double mutant. Specificity could also be shown by transfecting parasites with the mutated version of the Lac-gfp gene as shown in reference 42.

12. Figs 7 D and E need quantification of plaque sizes and statistical analysis

13. Figure 7F needs statistical analysis and the legend needs to indicate the number of biological replicates for G.

14. Figure 7H needs quantification and statistical analysis. Indicate the number of replicates. Could this part be re-organized with the legends at the top? Also, separate the bottom panel since it is very confusing as presented.

15. Figure 8: Legend needs to indicate the number of biological replicates. The data in B needs statistical analysis.

16. Figure 8: Authors show that DOC2.1 binding to PS liposomes is independent of calcium. The concentration of free calcium is difficult to predict from the information provided. EDTA mainly chelates Mg2+ and EGTA would be more appropriate. The concentration of Ca2+ of the lysate could be high considering that Ca2+ bound to proteins would be released during lysis of the cells. 0.1 and 1 mM EGTA should be tested to be sure that calcium is not relevant. In addition, there is no information of the Ca2+ binding affinity of DOC2.1

17. Abstract says that DOC2.1 senses changes in cytosolic Ca and that is a sensor of PS at the inner leaflet of the PM. The data only shows that mutating putative calcium binding domains results in growth defects due to egress defect. There is no evidence for Calcium binding of DOC2.1 so the mutation could impact other aspects of the protein. May be increases in cytosolic Ca affects the localization of DOC2.1 making it more defined? This would be interesting to explore.

18. The methods section does not have an explanation for the statistical analysis of the data.

**Part III – Minor Issues: Editorial and Data Presentation Modifications**

Reviewer #1: Line 33, insert ‘the’ parasite.

Line 64, insert ‘the’ plasma membrane.

Line 67, Perhaps modify ‘inner enriched gradient’ which doesn’t sound quite right. It reads well in line 133.

Line 68, delete ‘the’.

Line 73, 79 should be ‘phylum Apicomplexa’. Delete ‘of’.

Line 82, ‘stages’.

Line 89, define PKA as the cAMP-dependent protein kinase.

Line 90, delete ‘G’ so that PKG is defined correctly.

Line 96, Perhaps replace ‘system’ with ‘motor’

Line 117, ‘based on’.

Line 128, Perhaps replace ‘participating’ with ‘involved’ or ‘strictly participates’.

Line 136-139 This section containing two presumptions would benefit from a re-write and isn’t a strong end to the Introduction.

Line 137, ‘presumable’ should be replaced by e.g. ‘putative’.

Line 157, replace ‘aroused’ with ‘arisen’ 

Line 175, ‘focused’.

Line 177, e.g. delete ‘an’.

Line 199, e.g. delete ‘the’.

Line 200, replace ‘every’ with ‘each of’.

Line 201, delete ‘in repaired’.

Line 205, ‘parasites’.

Line 217, ‘of the complexes’.

Line 222, ‘cell’.

Line 224-225, please re-write the last half of this sentence.

Line 225, e.g. ‘A bulk time’.

Line 235, ‘parasites’.

Line 236, ‘the asymmetric PS distribution’.

Line 238, ‘taken together these results indicate that…’.

Line 242, ‘micronemes’.

Line 247, delete ‘the’.

Line 251, Delete ‘A’.

Line 293, specify that in P. yoelii it is GCβ

Line 294, ‘groups with’ would benefit from slight expansion.

Line 363, ‘remained’?

Line 629, state here what calcium salt was used.

Reviewer #2: Minor Notes:

• Line 157: please change “aroused" to “have arisen”

• Line 166: please consider changing sentence to “Similar to GC and its partner CDC50.1, which have previously been found at the apical tip of the parasites (8), ATP2A, ATP2B, CDC50.2 and CDC50.4 are also found at the apical tip”

• Line 191: why do some of the western blots have a parental sample while some don’t?

• Line 217: the authors provide no rationale behind the protease protection experiments. A short sentence here explaining the reasoning behind this or the line of thought would greatly aid the readers in following the story.

• Line 353: unclear what is meant by “tonic”

• The authors have size bars for all IFAs but they fail to state what size this bar corresponds to in any of the figure legends. Please include this

• The authors should consider merging Figures 7 & 8

• Figure 6: the scale bars of 6E should be amended to be the same as the scale bars the other figures

• Figure 6: the statistical analysis of Figure 6H is missing despite there being a p value

Reviewer #3: Minor issues:

1. Tables S1 and S2 are mislabeled.

2. The links in Table S2 (should be S1) all lead to a “page not found”

3. Figure 2C needs markers

3. Although is not relevant to the story, the authors mention in the introduction that the signaling cascade starts with cGMP, which activates PKG, PI-PLC, IP3, calcium and CDPKs. However, it should also be taken into consideration that PIPLC needs calcium for activity, so the pathway may not be that linear and calcium could also be upstream to PIPLC.

PLOS authors have the option to publish the peer review history of their article (what does this mean?). If published, this will include your full peer review and any attached files.

Reviewer #1: No

Reviewer #2: No

Reviewer #3: No
---

## [Decision Letter · Decision Letter 1]

22 Dec 2021

Dear Dominique,

Thank you very much for submitting your manuscript "Toxoplasma gondii phosphatidylserine flippase complex ATP2B-CDC50.4 critically participates in microneme exocytosis" for consideration at PLOS Pathogens. As with all papers reviewed by the journal, your manuscript was reviewed by members of the editorial board and by several independent reviewers. The reviewers appreciated the attention to an important topic. Based on the reviews, we are likely to accept this manuscript for publication, providing that you modify the manuscript according to the review recommendations.

As you will see, the revised version of this manuscript has been examined by 3 reviewers. The reviewers came to widely differing views, including a recommendation to reject the submission. After extensive consideration, the editorial view is that you should be given an opportunity to respond to these comments in a further round of revision. Whilst minor points of revision are raised by Reviewer #3, Reviewer #1 points out that in the absence of any data showing that DOC2.1 binds PS, there is now no obvious link between the DOC2.1 data in the revised manuscript and the data on the ATP2B/CDC50.4 complex. We agree that this substantially reduces the impact and breadth of the manuscript since it means that ther is now no experimentally validated functional link between DOC2.1 function and the function of the ATP2B/CDC50.4 complex. This conclusion is further strengthened by the fact that mutagenesis of DOC2.1 residues presumed to be involved in calcium binding affected egress but not invasion or microneme discharge. We would be grateful if you could address this major comment, as well as those of Reviewer #3. In addition, it is noted that the localisation by immunofluorescence of CDC50.4-mAID-HA in panels E and F of Figure 2 appears different from that of HA-tagged CDC50.4 (Fig 2B). Can you please address this issue too?

Sincerely,

Michael J Blackman

Associate Editor

PLOS Pathogens

Vern Carruthers

Section Editor

PLOS Pathogens

Kasturi Haldar

Editor-in-Chief

PLOS Pathogens

orcid.org/0000-0001-5065-158X

Michael Malim

Editor-in-Chief

PLOS Pathogens

orcid.org/0000-0002-7699-2064

As you will see, the revised version of this manuscript has been examined by 3 reviewers. The reviewers came to widely differing views, including a recommendation to reject the submission. After extensive consideration, the editorial view is that you should be given an opportunity to respond to these comments in a further round of revision. Whilst minor points of revision are raised by Reviewer #3, Reviewer #1 points out that in the absence of any data showing that DOC2.1 binds PS, there is now no obvious link between the DOC2.1 data in the revised manuscript and the data on the ATP2B/CDC50.4 complex. We agree that this substantially reduces the impact and breadth of the manuscript since it means that ther is now no experimentally validated functional link between DOC2.1 function and the function of the ATP2B/CDC50.4 complex. This conclusion is further strengthened by the fact that mutagenesis of DOC2.1 residues presumed to be involved in calcium binding affected egress but not invasion or microneme discharge. We would be grateful if you could address this major comment, as well as those of Reviewer #3. In addition, it is noted that the localisation by immunofluorescence of CDC50.4-mAID-HA in panels E and F of Figure 2 appears different from that of HA-tagged CDC50.4 (Fig 2B). Can you please address this issue too?

Reviewer Comments (if any, and for reference):

Reviewer's Responses to Questions

**Part I - Summary**

Reviewer #1: The authors have dealt with the reviewer comments comprehensively. It is good that they decided to pull back on the direct interaction of DOC2.1 and PS in light of further work being needed to substantiate this.

Reviewer #2: The revised manuscript by Bisio and colleagues incorporates several of the requested experiments and changes. However, some major comments were not addressed by the authors due to technical difficulties, even for experiments previously performed and published by the lab, such as assessing the involvement of the ATPases identified in flipping phospholipids other than PS. They also have not been able to reproduce consistently the DOC2.1 liposome binding experiments. As a result, the authors have removed a significant portion of the results section relating to DOC2.1. This is a very honest response, which I highly value, however, this leave the manuscript in a weaker position and slightly disjointed.

As the manuscript currently stands, the authors have convincingly shown that ATP2A and ATP2B bind to CDC50.4, and that ATP2B and CDC50.4 play an important role in egress, invasion and microneme secretion. They also show that knockdown of these two proteins leads to a block in PS flipping. While it is possible that ATP2B is directly involved in PS flipping, this has not been shown experimentally. The authors also show that ATP7B is important for lytic growth, but do not characterise this line further. Finally, the authors show that knockdown of DOC2.1, a protein thought to be involved in microneme secretion, leads to a block in lytic growth due to the inability of the parasites to egress, invade or secrete micronemes. By mutating the calcium binding sites of DOC2.1, they also show that these sites seem to be important for egress but not invasion or microneme secretion. This is a surprising result, but not further pursued. Since the authors removed the DOC2.1 liposome binding experiments, which showed binding of DOC2.1 to PS, there is no longer a link between the role of DOC2.1 and ATP2B/CDC50.4. This weakens the manuscript significantly. Another concern is that in Figure 2, the localisation of CDC50.4-HA in panel B appear to occupy a very different localisation compared to CDC50.4-mAID-HA in panel E. This seems to indicate that mAID-tagging of CDC50.4 mislocalises the protein.

Reviewer #3: This work studies the role of a complex formed between the flippase ATP2B with the cell division control protein (CDC) 50.4 for the flipping and uneven distribution of phosphatidylserine, which may act as lipid mediator for organelle fusion and microneme exocytosis. Authors demonstrated the interaction of CDC50.4 with ATP4B and focused the work on the function of these proteins. Downregulation of CDC50.4 led to degradation of ATP2B and the complex was critical for T. gondii growth. Invasion and egress was impaired in the mutants while intracellular replication was not. Microneme secretion of extracellular parasites was reduced upon downregulation of either ATP2B or CDC50.4. The authors claim that DOC2.1, a previously described key egress and invasion factor, senses changes in cytosolic calcium in intracellular parasites and may be the plausible sensor of PS at the inner leaflet of the plasma membrane.

This work is interesting, and it is an important contribution to our knowledge of the role of P4-ATPases flippases in T. gondii. Authors responded to most of the previous critique of this reviewer. Two minor concerns are indicated.

**Part II – Major Issues: Key Experiments Required for Acceptance**

Reviewer #1: (No Response)

Reviewer #2: (No Response)

Reviewer #3: No major issues

**Part III – Minor Issues: Editorial and Data Presentation Modifications**

Reviewer #1: (No Response)

Reviewer #2: (No Response)

Reviewer #3: Two minor concerns that could easily be clarified in the legends to the figures.

The legend for Figure 2C needs an explanation for how the westerns were done. It is mentioned the IPs with anti-Ty but the antibodies used for the western are not mentioned.

The PCRs presented in FigS4B are a little confusing. Could the authors include the size of the expected bands? There is a shadow/smear in part B. Does it mean anything? Please clarify what is expected and the result.

This figure is important to understand how the mutation of DOC2.1 was made.

PLOS authors have the option to publish the peer review history of their article (what does this mean?). If published, this will include your full peer review and any attached files.

Reviewer #1: No

Reviewer #2: No

Reviewer #3: No

Figure Files:

Data Requirements:

Reproducibility:

References:

---

## [Editor Report · Decision Letter 2]

11 Mar 2022

Dear Dominique,

We are pleased to inform you that your manuscript 'Toxoplasma gondii phosphatidylserine flippase complex ATP2B-CDC50.4 critically participates in microneme exocytosis' has been provisionally accepted for publication in PLOS Pathogens.

Best regards,

Michael J Blackman

Associate Editor

PLOS Pathogens

Vern Carruthers

Section Editor

PLOS Pathogens

Kasturi Haldar

Editor-in-Chief

PLOS Pathogens

orcid.org/0000-0001-5065-158X

Michael Malim

Editor-in-Chief

PLOS Pathogens

orcid.org/0000-0002-7699-2064

Dear Dominique

During the manuscript proofing stage, there are a small number of grammatical errors in the Abstract and Author summary that should be attended to, as follows:

Second sentence of Abstract: please alter 'micronemes' to 'microneme'

Penultimate sentence of Abstract: please delete 'the' from '...for the organelle fusion...'

3rd sentence of Author summary: replace 'depends' with 'depend'

4th sentence of Author summary: alter to '...act as a flippase...' (insert 'a')

Final sentence of Author summary: alter to '...the importance of membrane homeostasis...' (insert 'of')
---

## [Editor Report · Acceptance letter]

21 Mar 2022

Dear Dr Soldati-Favre,

We are delighted to inform you that your manuscript, "Toxoplasma gondii phosphatidylserine flippase complex ATP2B-CDC50.4 critically participates in microneme exocytosis," has been formally accepted for publication in PLOS Pathogens.

Best regards,

Kasturi Haldar

Editor-in-Chief

PLOS Pathogens

orcid.org/0000-0001-5065-158X

Michael Malim

Editor-in-Chief

PLOS Pathogens

orcid.org/0000-0002-7699-2064